# GH/IGF-1 Abnormalities and Muscle Impairment: From Basic Research to Clinical Practice

**DOI:** 10.3390/ijms22010415

**Published:** 2021-01-02

**Authors:** Betina Biagetti, Rafael Simó

**Affiliations:** Diabetes and Metabolism Research Unit, Vall d’Hebron Research Institute and CIBERDEM (ISCIII), Universidad Autónoma de Barcelona, 08193 Bellaterra, Spain

**Keywords:** acromegaly, myopathy, review, growth hormone, IGF-1

## Abstract

The impairment of skeletal muscle function is one of the most debilitating least understood co-morbidity that accompanies acromegaly (ACRO). Despite being one of the major determinants of these patients’ poor quality of life, there is limited evidence related to the underlying mechanisms and treatment options. Although growth hormone (GH) and insulin-like growth factor-1 (IGF-1) levels are associated, albeit not indisputable, with the presence and severity of ACRO myopathies the precise effects attributed to increased GH or IGF-1 levels are still unclear. Yet, cell lines and animal models can help us bridge these gaps. This review aims to describe the evidence regarding the role of GH and IGF-1 in muscle anabolism, from the basic to the clinical setting with special emphasis on ACRO. We also pinpoint future perspectives and research lines that should be considered for improving our knowledge in the field.

## 1. Introduction

Acromegaly (ACRO) is a rare chronic disfiguring and multisystem disease due to non-suppressible growth hormone (GH) over-secretion, commonly caused by a pituitary tumour [1]. The autonomous production of GH leads to an increase in the synthesis and secretion of insulin-like growth factor-1 (IGF-1), mainly by the liver, which results in somatic overgrowth, metabolic changes, and several comorbidities [1].

One of the significant disabling co-morbidities accompanying ACRO is myopathy and its related musculoskeletal symptoms [2,3], which are significant determinants of the low quality of life (QoL) of these patients and may persist despite disease remission [2,4,5,6].

However, we do not know the exact underlying mechanisms involved in its development. Although the duration of the disease and serum GH/IGF-1 levels were shown to be independent predictors of overall mortality and co-morbidities of ACRO [7,8,9,10,11], their specific role in the pathogenesis of myopathy remains to be elucidated. Besides, both GH and IGF-1 levels are not always directly and unequivocally associated with patient QoL and the presence and severity of co-morbidities [4,12]. For this reason, numerous efforts are currently being made to detect active disease, pointing to other variables apart from GH/IGF-1 [13,14].

Experimental research using cell lines and animal models is necessary to understand better the mechanisms involved in the myopathy induced by ACRO. In this regard, there is some evidence regarding the effects of the GH/IGF-1 axis in the muscular system and the mechanisms by which different conditions could impact on muscle performance.

This review aims to describe the evidence regarding the role of GH and IGF-1 in muscle anabolism, from the basic to the clinical setting, taking ACRO as the leading paradigm in the clinical setting. The new perspectives and scientific gaps in this field to be covered are also underlined.

## 2. GH/IGF-1 Axis in Humans

Pituitary synthesis and secretion of GH is stimulated by the episodic hypothalamic secretion of GH releasing factor and mainly inhabited by somatostatin [15]. GH stimulates the synthesis of IGF-1 mostly by the liver, and both circulating GH and IGF-1 inhibit GH secretion by a negative loop at both hypothalamic and pituitary levels. In addition, age, gender, pubertal status, food, exercise, fasting, insulin, sleep and body composition play important regulatory roles in the GH/IGF-1 axis [15].

### 2.1. GH Circulation, GH Receptor and Intracellular Signalling

In the circulation, GH binds to growth hormone binding protein (GHBP) which is a soluble truncated form of the extracellular domain of the GH receptor (GHR) [16]. Therefore, bound and free GH both exist in the circulation, and GH bioavailability depends on the pulsatile pattern of its secretion.

GHR belongs to the so-called “cytokine receptor family”, specifically part of the class I cytokine receptor family. Cytokine receptors lack intrinsic protein tyrosine kinase (PTK) activity and therefore rely on binding non-receptor PTKs for their signal transduction, the so-called Janus Kinase (JAK) (Figure 1). The dimerisation induced by the binding of GHR is the critical step in activating the signalling pathway JAK/STAT (Janus kinase/signal transducer and activator of transcription), and binding to GHR brings two JAK2 molecules close enough to initiate their trans-phosphorylation by recruiting STAT [17,18]. Phosphorylated STATSs, translocate to the nucleus, binding to specific DNA sequences and induce the transcription of specific GH-dependent genes, thus promoting not only the synthesis of IGF-1, IGF-1 binding protein (IGFBP) acid-labile sub-units (ALS), but also directly stimulating proliferative and metabolic actions (Figure 1).

GHR is widely distributed throughout the body. Although GHR expression is particularly important in the liver, GH receptors are found in other tissues like muscle, fat, heart, kidney, brain and pancreas [19]. The actions of GH depend not only on the receptor but also on the organ it stimulates; specifically, GH actions in the muscle are controversial and depend, among other factors, on GH quantity, the time of exposition and the condition of the subjects.

### 2.2. IGF-1 Circulation, IGF-1 Receptor and Intracellular Signalling

The majority of IGF-1 circulates in the serum as a complex with the insulin-like growth factor-binding protein IGFBP and ALS. The function of ALS is to prolong the half-life of the IGF-I-IGFBP- binary complexes, thus regulating IGF-1 bioactivity [20].

IGF-I receptors (IGF-1R) are highly homologous to the insulin receptor. For this reason, at high concentrations insulin cross-reacts with the IGF-1R and vice versa [21]. In cells expressing both insulin and IGF-I receptors as in the myocyte, the receptors undergo hetero-dimerisation (hybrid receptors), as well as homo-dimerisation. In skeletal muscle IGF-I receptors are mainly present as hybrids, together with an excess of classical insulin receptors regarding other mammalian tissue. Hybrid receptors bind both insulin and IGF-I, although they have a lower insulin affinity than classical insulin receptors [21,22].

In contrast to GHR, the IGF-1R is a member of the receptor tyrosine kinase (RTK) family. This type of receptor’s common characteristic is to have an extracellular domain for ligand binding, a single transmembrane domain and an intracellular domain with tyrosine kinase activity, which is activated after the ligand binds to the extracellular domain, thus triggering the phosphorylation of Insulin Receptor Substrates (IRS). Once the IRS is phosphorylated, it activates different pathways such as phosphatidylinositol 3-kinase or RAS (Figure 2).

The number of IGF-1Rs in each cell is tightly regulated by several systemic and tissue factors, including circulating GH, thyroxine, platelet-derived growth factor, and fibroblast growth factor. Therefore, in vivo, the number of receptors alone is not the only determining factor in IGF-1 function [20] (Figure 2).

Despite the ubiquity of IGF-1R, the in vitro biological effects of IGF-1 are relatively weak and often are not demonstrable except in the presence of other hormones or growth factors, thus suggesting that IGF-1 acts as a permissive factor augment the signals of other factors [22]. Furthermore, IGF-1 and insulin share common signalling pathways and functional overlap. However, IGF-1 exerts a more decisive action than insulin in activating the RAS/MAP kinase pathway, which is related to cell growth, protein synthesis, and apoptosis. By contrast, insulin (and in lower grade IGF-1) regulates the metabolism of carbohydrates, lipids and proteins via PI-3K/Akt, [20,22] (Figure 2).

## 3. The Skeletal Muscle: A Central Link between Structural and Metabolic Function

The skeletal muscle is one of the three significant muscle tissues in the human body, along with cardiac and smooth muscles [23]. It is crossed with a regular red and white line pattern, giving the muscle a distinctive striated appearance, leading to it being called striated muscle. There are three types of muscle fibres, which differ in the composition of contractile proteins, oxidative capacity, and substrate preference for ATP production. Type I fibres or slow-twitch fibres have low fatigability (better for endurance), and display slow oxidative ATP, primarily from oxidative phosphorylation (a preference for fatty acids as substrate) [24,25]. Type II or fast-twitch fibres have the highest fatigability, the highest contraction strength, while glucose is the preferred substrate [24]. They are further divided into Type IIa and IIb fibres. Type IIa is used for intermediate endurance contraction and, in general, relies on aerobic/oxidative metabolism. Both types I and IIa fibres are considered red fibres because of their high amounts of myoglobin. They also contain high numbers of mitochondria, Type IIb fibres, (also called IIX/d) or fast glycolytic fibres, are the fastest twitching fibres and are more insulin resistant [26]. They are also white fibres with low levels of myoglobin and a high concentration of glycolytic enzymes and glycogen stores. This is because they produce ATP primarily from anaerobic/glycolysis [27,28]. Thus, the fibre type composition of skeletal muscle impacts systemic energy consumption and vice versa. Endurance or aerobic exercise increases mechanical and metabolic demand on skeletal muscle, resulting in a switch from a fast-twitch to a slow-twitch fibre type [24].

In summary, skeletal muscle’s structural function is the most obvious one. It gives structural support, helps maintain the body’s posture, and exerts the muscle contraction leading to movements. Furthermore, skeletal muscle is also an important site of the intermediary metabolism: it is a storage site for amino acids, plays a central role in maintaining thermogenesis, acts as an energy source during starvation and, along with the liver and adipose tissue, is crucial in the development of insulin resistance in type 2 diabetes mellitus [29].

## 4. GH/IGF-1 Actions in the Muscular System

The muscle is a primary target of GH and IGF-1 [30,31]. The final contribution of GH and IGF-1 in any effect on the muscular system is not easy to establish in-vivo. Through its receptors, GH action can be both direct and indirect through the induced production of IGF-I. Furthermore, IGF-I is also produced locally in tissues and can act in a paracrine/autocrine manner [32,33].

As we commented above, the muscle is much more than an organ of support and contraction; it is a main metabolic organ. The fibre type composition of skeletal muscle impacts systemic energy consumption and vice versa [24]. Both hormones act in the muscle, modifying its metabolic function and structure. For example, Bramnert et al. (2003) [34] studied both the short-term (1 week) and long-term (6 months) effects of a low-dose (9.6 μg/kg body weight/d) GH replacement therapy or placebo on whole-body glucose and lipid metabolism and muscle composition in 19 GH-deficient adult subjects. GH therapy resulted in glucose metabolism deterioration and an enhanced lipid oxidation rate in both short and long-term treatment groups, reflecting a fuel use switch from glucose to lipids.

Furthermore, there was a shift toward more insulin-resistant type II X fibres in the biopsied muscles during GH therapy. These studies significantly increased our knowledge regarding the relationship between skeletal muscle and metabolism’s fibre type composition. However, whether this muscular fibre shift represents the cause or the consequence of insulin resistance requires further investigation.

Moreover, GH/IGF1 levels have been related to ageing. In particular, IGF-1 production by the skeletal muscle is impaired in older humans and linked with aged related sarcopenia [30]. Therefore, GH/IGF-1 actions in the muscle have structural, metabolic and ageing consequences and vice versa.

### 4.1. GH

The first reports on the anabolic growth hormone effects dated back to 1948 and suggested that GH inhibits proteolysis during fasting [35]. Subsequent studies have expanded this concept [36,37].

GH is the primary hormone in the fast state; from an evolutionary point of view, when food is scarce, GH changes fuel consumption from carbohydrates and protein to the use of lipids, allowing for the conservation of vital protein stores [38]. Humans’ nitrogen balance must be kept stable to maintain the pool of essential amino acids (EAA) involved in protein formation. Considering this balance, different EAA radiolabelling techniques have been used to evaluate the effects of GH and IGF-1 on protein metabolism. Table 1 summarises the results of studies assessing protein anabolism using radioisotopes. The action of GH on muscle tissue is mainly anabolic and has little effect on proteolysis. Still, the effects could be different depending on the circumstances, dose and time of exposition. The acute infusion of GH on amino acid metabolism results in a profound reduction in amino acid oxidation [36,37,39,40,41]. Some studies demonstrated that acute GH infusion increased whole-body protein synthesis without any significant changes in the synthetic rate of muscle proteins. By contrast, other studies using higher doses of GH found lower proteolysis rates. In general, acute GH infusion exerted an anabolic effect on whole-body amino-acid (AA) metabolism, evidenced in reduced leucine oxidation, increased nonoxidative leucine disposal, an increased amino acid disappearance rate (Rd) (all indexes of protein synthesis) and less effect on protein breakdown (a lower rate of amino acid appearance (Ra)) probably related with dosage. Regarding chronic GH exposure, it seems more consistent than acute administration in promoting an increase in whole-body protein synthesis [42,43,44]. Similar results have been reported after GH administration in patients with growth hormone deficiency (GHD) [45,46], and after GH administration in malnourished patients under haemodialysis [47].

Norreland et al. (2001) [48] evaluated the action of GH on fasting longer than overnight. They found that GH suppression led to increased proteolysis and with GH replacement they found a significant decrease in branched-chain amino acid levels, consistent with decreased proteolysis.

Nevertheless, in studies designed to recover the muscle loss in the elderly, the overall impact of GH administration has been far lower than expected [44,49,50]. Little is known about GH uses in fitness [51]. However, recently, in patients in need of reparative knee surgery Mendias et al. (2020) [52] found that rhGH compared to placebo improved quadriceps strength and reduced MMP3 (a subrogated marker of cartilage degradation).

Altogether, these experiments support the anabolic action of GH with less effect on proteolysis, depending on time length and the context of GH exposition. These effects probably become more evident with more prolonged GH exposure or in a pathological state as acromegaly accompanied by elevated insulin and IGF-I levels.

### 4.2. IGF-1

IGF-1 is one of the most important signals in muscle anabolism and repairs [30,53,54]. As mentioned above, IGF-1 is secreted by the liver after GH stimuli and directly from skeletal muscle in an autocrine/paracrine manner [33]. IGF-1 mediates beneficial outcomes of physical activity [55] and can prevent chronic disease [56].

The first reports of IGF-1 date back to 1957 as “sulphation factor”, when Salomon et al. described its ability to stimulate 35-sulphate incorporation into rat cartilage. Later, IGF-1 (called somatomedin-C) was assumed to be a mediator of anabolic and mitogenic GH activity [57], but this concept drastically changed after verification of the local production and action of IGF-1 [33,58].

Immediately after IGF-1 biosynthesis [59], a new challenge in clinical research was opened up. First, Guler et al. (1987) [60] compared eight healthy adult volunteers, the short-term metabolic effects of IFG-1 and insulin. Next, Laron et al. (1990) [61,62] investigated IGF-1 impact on GHRH in healthy adults and children with primary IGF-1 deficiency (GH insensitivity) or Laron’s Syndrome (LS). Laron et al. were also the first to introduce the long term administration of biosynthetic IGF-1 in LS [63,64]. Some of the most relevant studies in humans, with systemic IGF-1 administration and its impact on protein and the muscular system, is summarised in Table 2.

In healthy people, Clemmons et al. (1992) [65] found evidence that short term IGF-1 infusion, was able to reverse fasting diet-induced muscular catabolism. There is also some evidence indicating that the effects of the systemic administration of IGF-I on protein metabolism depend on environmental AA levels. In this regard, Turkalj et al. (1992) [66] showed that IGF-I administered without AA, suppressed proteolysis, while in the study by Russell-Jones DL, et al. (1994) [67]. IGF-1 administration in combination with AA infusion increased protein synthesis. Likewise, Kupfer et al. (1993) [68] found that GH and IGF-1 anabolic effects were enhanced when co-administered in 7 healthy calorically restricted patients. Lagger et al. (1993) [69], compared high and low doses of rhIGF-1 and found that high doses of IGF-1 decreased proteolysis more than insulin. However, it was accompanied by an unexpected inhibition of protein synthesis.

More recently, some researchers have studied the effects of rhIGF-1 on muscle performance in special populations. Rutter et al. (2020) [70] investigated 21 boys with Duchenne muscular dystrophy and found that after 6 months of rhIGF-1 therapy, despite improvements in linear growth, muscular motor function did not change. Guha et al. (2020) [71] studied fifty-six recreational athletes and observed that muscular aerobic performance improved after 28 days of rhIGF-1/rhIGFBP-3 administration, but no effects on body composition. It should be taken into account that circulating IGF-1 is probably less important in those tissues that can produce the hormone themselves, such as skeletal muscle [72]

Focusing on ACRO, some degree of IGF-I receptor resistance to IGF1 action has recently been reported [73]. This important finding could explain why in the presence of high IGF1 levels, the metabolic GH effects are predominant in these patients. In addition, IGF-1 resistance might also be involved in other ACRO related muscle disturbances.

In summary, these data suggest that the administration of IGF-1 inhibits proteolysis and improves muscle performance with less effect on protein synthesis. Other factors such as local IGF-1 secretion and/or IGF1R resistance could play an important role. The data is still controversial, and the combined and additive effect of IGF-1 and GH co-administration remains a subject of debate.

## 5. The Key of mTOR in Muscle Structural and Metabolic Function

Skeletal muscle mass and whole-body metabolism are closely related. One of the major players in coordinating growth with nutrient availability is rapamycin’s mammalian target (mTOR) [74]. There are two different mTOR complexes, mTORC1 and mTORC2. In brief, mTORC1 senses nutrients, oxygen, energy, and growth factors promote cell growth, regulates metabolic processes, and mTORC2 responds more specifically to growth factors and regulates cell survival and metabolism [75]. Specifically, mTORC1 plays a key role in muscle structure, stimulating postprandial and post-exercise muscle protein synthesis. This pathway is mainly activated by mechanical contraction (i.e., exercise), which causes paracrine IGF-1 secretion [72]. However, AA intake also participates in governing the mTORC1 pathway. It has been shown that leucine is a potent stimulator of muscle protein synthesis by triggering mTORC1 signalling in human skeletal muscle [76]. Recent work has found that during AA sufficiency, mTORC1 kinase activity is stimulated mainly in 3 ways: (1) The lysosomal AA transporter SLC38A9 sensing arginine within the lysosome [77]; (2) The leucine sensor Sestrin2 [78]; (3) The arginine sensor [cellular arginine sensor for mTORC1 (CASTOR1)] [79]. All these pathways ultimately allow mTORC1 to be activated (Figure 3).

Similarly to IGF-1R resistance described in ACRO, an “anabolic resistance” due to an impairment of the “sensor” capacity of the muscle to metabolise circulating AA has been related to the sarcopenia that occurs in the elderly [76]. This reduced ability is probably associated with impairments in molecular signalling such as decreased amino acid transport, which results in reduced mTORC activation.

Recently, we have reported low circulating levels of branched-chain amino acid in active ACRO [80]. This finding could contribute to the myopathy observed in patients with ACRO, but further basic and clinical research on this issue is needed. In this regard, to the best of our knowledge, there are no studies addressed to investigate the potential role on muscle performance of diet enriched with AA in ACRO.

## 6. Impact of GH/IGF-1 Axis Impairment on Muscle Metabolism and Function: Lessons from Basic Research

The GH and IGF-1-mediated regulation of skeletal muscle structure and performance have recently been reviewed [54,81,82]. We will focus on the impact on the muscle of diseases with low or excess GH/IGF-1.

### 6.1. Primary Cell Cultures

In vitro studies using immortalised the mouse myoblast cell line C2C12 has provided preliminary evidence regarding the action of IGF-I in muscle stimulating muscle protein synthesis, and also in suppressing proteolysis [83]. IGF-1 is also related to myoblast proliferation [84] and differentiation [54]. In addition, IGF-1 modulates the expression of L-type amino acid transporters in the muscles of spontaneous dwarf rats [85].

Mahendra et al. (2010) [86], in an elegant work using cell lines of mice lacking either the GH receptor or the IGF-1 receptor in skeletal muscle, found that GH mediates skeletal muscle development by enhancing myoblast fusion in an IGF-1-dependent manner. They showed that GH treatment quickly increases IGF-1 mRNA and that this autocrine production of IGF-I leads to a significant increase in primary myoblast proliferation. By contrast, disruption of the GH receptor in skeletal muscle produces marked alterations in muscle nutrient uptake and insulin sensitivity in an IGF-1-independent manner.

Altogether, these findings indicate that the myoblast anabolic process is a complex action where both hormones are complementary. Nevertheless, metabolic dysfunctions seem to be more related to GH impairment.

### 6.2. Transgenic Mice

Several genetically modified animal models (giants and dwarfs) have been generated and characterised in the last 30 years [87,88,89] and have been used to investigate the phenotype, consequences and mechanisms underlying the altered GH/IGF-1 axis [90]

This review will focus on the most relevant ones: transgenic mice overexpressing GH and GH knockout mice. For comprehensive reviews, please see references [87,91,92,93].

High GH levels in transgenic mice have elevated GH and IGF-1 levels, increased body weight, and lean mass. The extreme elevation in circulating GH also results in hyperinsulinemia despite euglycemia, probably due to reduced insulin receptors or GLUT4 channels in skeletal muscle. There are some differences among the various GH transgenes species in mice (e.g., ovine GH, bovine GH or human GH). Human GH can bind to both GH and prolactin (PRL) receptors, whereas rat, bovine and ovine GH bind exclusively to GHR. Thus, overexpression of human GH in mice results in a physiological state in which both GH and PRL receptors are simultaneously activated in contrast with rat, bovine, or ovine where only GHR is activated [92]. Together with the different ages of the mice used in the experiments, these differences could explain the heterogeneity of the published results regarding body composition. Besides, the following aspects should be taken into account when using transgenic mice: (1) In ACRO, GH secretion is from a pituitary adenoma whereas, in GH transgenic mice, GH excess is due to transgene insertion resulting in ectopic GH secretion. (2) The level of circulating GH is often markedly greater in transgenic mice than is observed in ACRO. (3) Transgenic mice exposure to excessive GH begun immediately after birth mimics gigantism more than ACRO (4) in human transgenic mice, GH and PRL receptors are simultaneously activated, unlike in ACRO.

In complete contrast to high GH level transgenic mice, there are the dwarfism models: The GH knockout mice (GHKO or GH−/−), completely lack GH due to a targeted deletion of the GH gene. These mice represent a model of human isolated GH deficiency. The GH receptor gene disrupted mice (GHR−/−) are mice completely resistant to GH action; they also have a pronounced decrease in body size with extremely low circulating IGF-1 despite elevated GH levels and constitute the model for LS. Other types of dwarfism models are transgenic IGF-1 knockout mice. Although the animals are born alive, mortality rates range from 32% to 95% due to respiratory failure caused by impaired diaphragm development, severe muscle dystrophy and intercostal muscles. These mice also develop metabolic abnormalities such as mild hyperglycaemia (250 mg/dl) and decreased pancreatic beta cells [63,94].

Finally, another model of IGF-1 deficiency is a consequence of a deletion of the gene encoding the IGF-1 in the liver (only in the liver, not in other tissues), resulting in a 75% reduction of circulating plasma IGF-1 levels [95]. These animals usually grow and are fertile. It has been suggested that high levels of growth hormone and increased paracrine or tissue IGF-1 levels compensate for the lower plasma levels of IGF-1.

Overall, these findings provide evidence that both GH and IGF-1, including locally muscular secreted IGF-1, have active implications for growth and muscle development.

### 6.3. Experimental Research Focused on the GH/IGF-1 System Effects on Muscle

The effects of GH deficiency in muscle have been controversial. Ayling et al. (1989) [96] reported a 50% reduction in type I fibres after rat hypophysectomy, that recovered after GH therapy. Similar results were obtained by Loughna and Bates five years later (1994) [97]. By contrast, Yamaguchi et al. (1996) [98] also observed a significant increase in type I fibres after rat hypophysectomy. Some studies have also reported no change in the composition of type I or type II fibres after GH replacement in the same experimental model [99]. These divergent results could be explained by the different exposure to GHD, GH therapy duration, and the associated deficit of other hormones such as thyroxine, testosterone and glucocorticoids.

Nielsen et al. (2014) [100] in an elegant work, measured the ultra-structure and collagen in the Achilles tendon, employing three groups of mice: (1) Giant transgenic mice that expressed bovine GH (bGH): the ACRO model; (2) Dwarf mice disrupted GH receptor GHR−/− (the LS model) and (3) A wild-type control group. They found that the mean collagen fibril diameter was significantly decreased in the ACRO and LS models. The ultra-structural pattern was more severely affected in the GHR−/− mouse model.

Similarly, a recent study of Cinaforlini et al. (2020) [101] investigated muscle repair and found that GH administration accelerated the muscle healing process.

Regarding the direct effect of IGF-1 on the muscle, Schoenle et al. in 1982 [102] reported that IGF-1 stimulated growth in hypophysectomised rats, thus providing evidence of independent IGF-1 action from GH. DeVol et al. in 1990 described for the first time the local skeletal muscle production of IGF-1 [58]. Using a rat model of compensatory hypertrophy of the soleus, the authors showed that muscle hypertrophy and local IGF-1 production occurred independently of GH. However, compensatory hypertrophy was not blunted in hypophysectomised rats.

All this evidence reinforces the concept that although the actions of GH and IGF1 in the muscle, are independent, they are complementary.

## 7. The Muscle in GH/IGF1 Axis Deficiency

Isolated GH deficiency (IGHD) either congenital or acquired, is the commonest pituitary hormone deficiency [103]. First description and treatment date from 1963 [104]. During childhood and puberty, the classical phenotype of GHD is short stature with height ≤ −2 SDS, frontal bossing, mid-facial hypoplasia (“doll-like” facies) and truncal adiposity [105]. Adult-onset patients with GHD have increased total and visceral fat, low bone mass, reduced muscle strength and impaired anaerobic physical capacity, an unfavourable cardiovascular profile, and low quality of life [106,107].

There is conflicting data regarding body composition and muscle structure and performance after GH treatment. In a recent study, Andrade-Guimarães et al. (2019) [108] found that individuals with congenital untreated IGHD had better muscle strength parameters adjusted for weight and fat-free mass than controls, and also exhibited greater peripheral resistance to fatigue.

In adults with GHD, Díez et al. (2018) [109] and Jørgensen et al. (2018) [107] among others, argued in favour of the beneficial effects of GH administration on body composition, muscle exercise capacity, bone structure and serum lipids. By contrast, He and Barkan (2020) [110] critically reviewed the evidence of treatment benefits in adult-onset GHD. They found most of the data inconsistent, thus recommending that GH treatment should be a personalised decision.

Regarding GH resistance (LS) syndrome, the main clinical characteristics are dwarfism, specific facial features such as a protruding forehead in the presence of a subnormal head circumference, sparse hair, skeletal and muscular underdevelopment, obesity and underdeveloped genitalia [111]. Long-term observations have shown that these individuals are protected from cancer [112,113]. In relation to the muscle, they have a reduced lean body mass and reduced muscle force and endurance [114]. Curiously, growth velocity obtained with IGF-I administration is smaller than that observed with hGH in children with congenital isolated GH deficiency [115]. Muscle benefit with IGF-1 therapy has not been fully established. Long-term treatment for adult patients living with LS aimed at strengthening the muscular and skeletal systems is not currently approved.

## 8. Muscle Impairment in Human Acromegaly

Muscular weakness and pain represent one of the significant disabling co-morbidities in ACRO [5,116] and, as above mentioned, despite long-term remission of the disease, this co-morbidity may persist and deeply compromise QoL. Although musculoskeletal pain is reported in up to 90%, it is not usually recorded in the clinical history [116]. It should be noted that as a result of treatment improvement, life expectancy in people living with ACRO is currently comparable to that of the general population [117] and, therefore, the rate of musculoskeletal impairment is not expected to be reduced. Nevertheless, this is one of the least studied co-morbidities [118], and there are few studies focused on the evolution of myopathy after ACRO treatment.

Mastaglia et al. in 1970 [119] reported for the first time on muscle impairment in ACRO. Mild proximal muscle weakness was present in six out of 11 cases. Histological changes in hypertrophy of both type-I and type-II muscle fibres were current in five out of nine biopsy specimens taken from a proximal limb muscle. However, electron microscopy showed a patchy myopathy process, thus accounting for these patients’ weakness and myalgia. By contrast, Nagulesparen et al. (1976) [120] studied 18 patients with acromegaly of varying degrees of severity; half of the specimens showed hypertrophy of type 1 fibres, and atrophy in type 2 fibres, although a direct correlation between muscle appearances and growth hormone levels was not observed.

These findings suggest, firstly, a dissociation between muscle macro/microstructure and the IGF-1/GH level, and secondly an alteration in the damage repair mechanism in acromegaly. Thus, other factors independently of hormone levels could play a key role in ACRO myopathy, and treatments strengthening the muscle damage repair mechanism could improve this co-morbidity.

Concerning energy consumption, Szendroedi et al. (2008) [121], found that patients with ACRO exhibited reduced muscular ATP synthesis and oxidative capacity that may persist despite the normalisation of GH secretion. Regarding strength, Füchtbauer et al. (2017) [5] studied ACRO patients at diagnosis (n = 48), one year after surgery (n = 29) and after long-term follow-up (median 11 years) (n = 24), compared to healthy subjects. They found reduced grip and strength in active ACRO and increased proximal muscle fatigue even after GH/IGF-1 normalisation.

Although the most apparent muscular structural change in active ACRO seemed to be hypertrophy [122,123], more recent studies, have not confirmed this finding [124]. Recently Gokce et al. (2020) [125] using ultrasound found that the thicknesses of many components of the quadriceps muscle were lower in ACRO patients than control subjects matched by age, sex and the body mass index.

We have very little evidence on interventional programs to improve this devastating co-morbidity. However, Lima et al. (2019) [6] evaluated the effects of a functional home rehabilitation program in seventeen adults with ACRO that resulted in improvements in muscle function, functional capacity, general fatigue, body balance, and QoL.

Accordingly, other therapeutic strategies apart from GH/IGF-1 normalisation, with a direct impact on muscle structure/performance should be followed to improve our patients QoL.

## 9. Concluding Remarks and Future Perspectives

In ACRO, the sustained increase of GH and IGF-1 critically affects the skeletal, muscular system with severe repercussions on life quality. At present, our knowledge regarding the underlying mechanisms and outcome of different treatments is scarce.

In addition, there are no clinical guidelines or any structured recommendations for dealing with this debilitating co-morbidity. Therefore, there is a pressing need to promote interaction between basic and clinical research, improving our knowledge and permitting us to design therapeutic strategies.

Nevertheless, we can remark on some important points: (1) Up to 90% of ACRO patients suffer from musculoskeletal pain. (2) This co-morbidity may persist despite biochemical control of the disease and deeply compromise QoL. (3) Other factors apart from GH/IGF-1 levels could have a significant role in ACRO myopathy’s underlying mechanism. (4) New biomarkers linking muscle mass and metabolism to diagnose better and phenotype myopathy and design the best therapeutic strategy are needed.

Cell lines and transgenic mice give important insights into the many actions of GH and IGF-1. The degree of the contribution of GH or IGF-1 to myopathy in acromegaly remains to be further investigated. In the clinical setting, we need (1) tools that efficiently and reliably permit us to evaluate the muscular status of patients with ACRO, (2) collaborative, multicentre studies with larger samples to provide more accurate data, and (3) further studies that investigate treatments that directly influence muscle performance, even in patients in remission.

Finally, it should be emphasised that the muscle should be seen as a key role organ, with mechanical and metabolic functions. Disturbances in metabolic processes (for example, hyperglycaemia, amino acid deficiency) can ultimately impact muscle structure and, vice versa, changes in fibre type can promote metabolic changes. In this regard, studies aimed at assessing the molecular link between structural changes and metabolic function are needed.

## Figures and Tables

**Figure 1 ijms-22-00415-f001:**
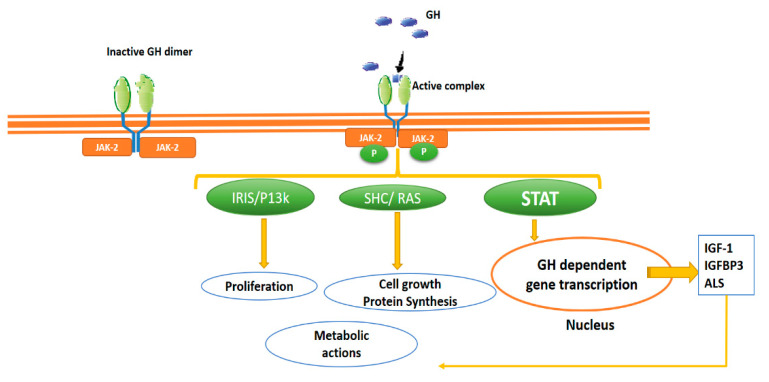
Growth hormone (GH) receptor. The GH receptor’s dimerisation is the critical step in activating the signalling pathway JAK/STAT. Phosphorylated STATS translocate to the nucleus, inducing the transcription of specific GH-dependent genes, promoting mainly the synthesis of IGF-1, IGFBP acid-labile sub-units, but also, stimulating other pathways (i.e., IRIS and SHC) that regulate cellular proliferation, growth and metabolic events. In addition, newly synthesised IGF-1 promotes metabolic and cellular events. IGF-1: insulin-like growth factor, GH: growth hormone IGFBP3: IGF-1 binding protein type 3, JAK/STAT: Janus kinase/signal transducer and transcription activator IRS: insulin receptor substrate; PI3K: phosphatidylinositol 3-kinase; SHC: (adaptor proteins).

**Figure 2 ijms-22-00415-f002:**
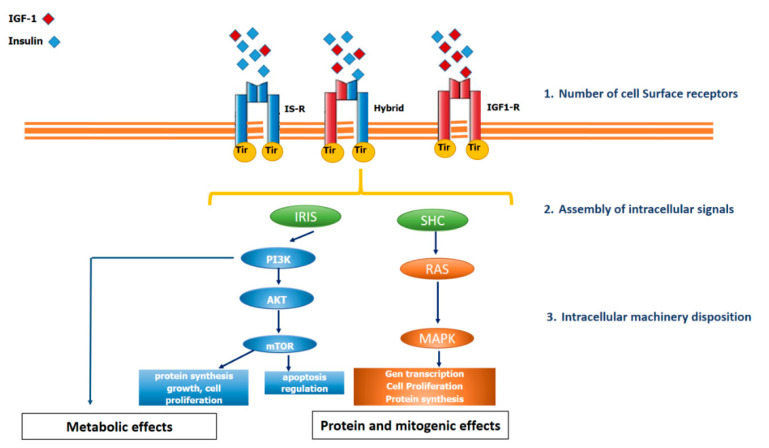
IGF-1 receptor, insulin receptor and hybrid receptor. The insulin receptor (IS-R), the insulin-like growth factor receptor type I (IGF-1-R) and the hybrid receptor belong to the family of receptor tyrosine kinases. The figure shows a certain degree of functional overlap. After binding ligands, the receptors stimulate the phosphorylation of tyrosine residues, which then induces phosphorylation of the Insulin Receptor Substrate (IRS) and SHC, which dock proteins for activating the phosphatidylinositol 3-kinase (PI3K-Akt) pathway or the Ras/MAPK pathway, respectively. The PI3K-Akt pathway is predominantly involved in metabolic actions (glucose uptake, insulin sensitivity) and cell proliferation, whereas the RAS/MAPK pathway, is primarily involved in mitogenic effects (proliferation, growth). The number of receptors alone is not the only determining factor in IGF-1 function. In this regard, the assembly and intracellular machinery disposition are also involved.

**Figure 3 ijms-22-00415-f003:**
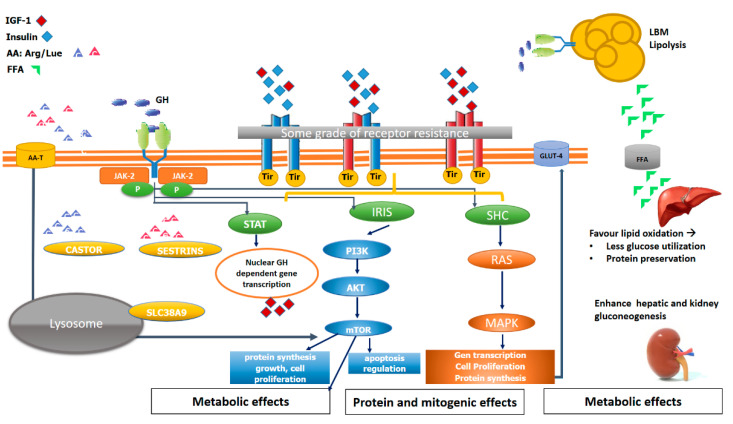
The interplay of GH, IGF-1, insulin, hybrid receptor and nutrients in the muscle. Schematic and simplified signalling pathway and the interplay of GH, IGF-1, insulin, hybrid receptor and nutrients in the muscle. AA (amino-acid) stimulates mTOR kinase activity mainly in 3 ways: (1) The leucine sensor Sestrin, (2) the lysosomal sensor SLC38A9, (3) The arginine sensor [cellular arginine sensor for mTORC1 (CASTOR1)]. GH activates via IRIS and SHC protein and mitogenic effects as well as IGF-1R/InsulinR and HybridR. GH also stimulates via JAK/STAT local muscular IGF-1 secretion. In addition, the metabolic effects of GH impact on muscle anabolism. The main metabolic action of GH is a lipolytic action on adipose tissue, releasing free fatty acids (FFA), which favours protein preservation, as well as liver and kidney-enhanced gluconeogenesis.

**Table 1 ijms-22-00415-t001:** Most relevant studies with GH evaluating protein anabolism using radioisotopes.

Author	Subjects	Design	GH Dose	Effects	Conclusions
**Fryberg**[39]	7 healthy	Brachial artery infusion no placebo	0.014 μg/kg/minTo rise locally notsystemic	Rd PHe 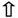 BCAAs 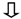 release	Locally infused GH stimulates skeletal muscle protein synthesis.
**Fryberg**[40]	7 healthy	Brachial artery infusion 3 h GH and then 3 h GH + insulin/no placebo	GH 0.014 μg/kg/minInsulin 0.02 mU/kg/minTo rise locally not systemic	3 h Rd 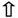 6 h Rd 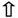 Ra 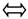 BCAAs 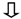	GH blunted the action of insulinto suppress proteolysis
**Yarasheski**[44]	18 healthyman	Resistance training GH/placebo	GH 40 μg/dfor 12 weeks	protein synthesis 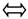 Leu. Ox 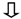	Similar muscle size, strength and protein synthesis
**Fryburt and Barret**[36]	8 healthy	Systemic brachial GH infusion/no placebo	0.06 μg/kg/min for 6 hsystemic IGF-1 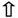	Rd 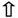 Ra 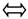	Acute stimulation of muscle but not whole-body protein synthesis
**Russell-Jones**[45]	18 adults GHD	Double-blind, placebo-controlled trial	0.018 IU/kg/day for 1 month followed by 0.036 IU/kg/day for 1 month	Leu Ra in either the placebo or GH-treated 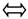 In GH groupLeu Rd 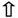	GH action in adults with GHD is due to an increase in protein synthesis.
**Copeland**[37]	15 healthy	Infusion/SSA +/− GH/control	2 μg/kg/hfor 3.5 h	Ra 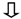 leu Ox 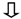	Acute GH did not affect muscle protein synthesis despite enhanced protein synthesis in non-muscle tissue
**Garibotto**[47]	6 Dialysis	Prospective cross over trial 6-week run and 6-week washout/no control	5 mg 3 times a week for 6 weeks.	Rd 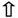 BCAAs 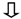	Increased muscle protein synthesis anddecreased negative muscle protein balance.
**Norrelund**[48]	8 healthy basal or after 40 h fasting	Systemic infusion/pancreatic clamp/controlled	4.5 IU(1) basal(2) after 40 h of fasting(3) after 40 h of fasting + SSAs(4) after 40 h of fasting with SSAs + GH replacement.	GH suppressionRa 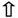 -----------GH replacementRa 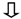 BCAAs 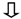	Suppression of GH during fasting leads to a 50% increase in urea-nitrogen excretion and increased proteolysis.
**Nielsen**[46]	7 GHD	Subcutaneous GH/controlled	GH replacement+/− acipimox	Ra 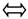 Rd 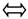	Lowering FFA with Acipimox increased whole body and forearm protein breakdown, and protein synthesis
**Short**[43]	9 healthy	Randomised crossover design GH/FS	GH (150 g/h; 2,1 ± 0.1 g/kg h)	Ra 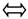 Rd 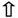	After GH synthetic rate of muscle proteins 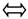
**Buijs**[41]	6 NW6 OB	Crossover designplacebo	A 1-h iv infusion of SSAs and GH (12 mU/kg/h or placebo	Leu Ox 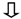	Administration of GH blunted the rise in Leu Ox similarly in both NW and OB
**Gibney**[42]	12 hypopit. man	Open-label randomised crossovertestosterone and GH	0.5 mg daily 6 weeks		No effect in protein breakdown, suppressing protein ox and stimulating protein synthesis

Adapted from: Jørgensen JOL, Christiansen JS. Growth Hormone Deficiency in Adults. Book. Karger Medical and Sci-entific Publishers; 2005. 
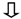
 Decrease 
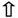
 Increase, 
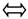
 not change. Ins: insulin, Leu: leucine, Ox: oxidation, BCAAs: branched-chain amino acids, Phe: phenylalanine, NW: normal weight, OB: obese, wk week. Units: Kg: kilogram, h: hours, IU: international units, min: minutes, mg: milligram, mU: primary units, μg: microgram. Rd: (Phe/Leu disappearance rate) represents protein synthesis. Ra: (rate of appearance of Phe/Leu) represents proteolysis. Hypopit: hypopituitarism. Ox—oxidation.

**Table 2 ijms-22-00415-t002:** Most relevant studies in humans, with systemic IGF-1 and muscle actions.

Author	Subjects	Design	Dose	Muscle Anabolism	Muscle Function	Conclusions
**Clemmons**[65]	6 healthy	2 weeks calorically restricted	rhGH: 0.05 mg/kg *6 d or iv infusion ofrhIGF-I 12 μg/kg IBW *16 h	With IGF-1 serum urea nitrogenthe reduction was greater than with GH	NA	IGF-1 reversed the catabolism caused by a restrictionof the diet.
**Turkalj**[66]	19 healthy man	Acute randomised rhIGF-1 ascending doses vs. saline.	IGF-I doses5, 7.5, 15 and 30 μg/kg/h(n = 4) and saline control (n = 3).	Dose-dependent decreaseof leucine oxidation.	NA	GF-1 proteolysis 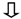 IGF-1 protein synthesis 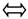
**Russell-Jones**[67]	5 healthyadequate substrate supply condition),	Random order of both IGF-1 or insulin 1 week apart + AA infusion	3-h IV infusion of IGF-1: 43.7 pmol*kg/min or Insulin 3.4 pmol/kg/min	IGF-1 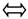 Ra 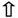 RdInsulin 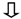 Ra 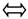 Rd	NA	IGF-1 increases protein synthesis in contrast to insulin,which acts to reduce proteolysis.
**Kupfer**[68]	7 healthy	2 weeks calorically restricted(20 kcal/kg IBW per d), with 1 g protein/kg IBW.	IV infusion ofrhIGF-1 12 μg/kg IBW *16 h *5 dorrhIGF-1 (same doses above) + rhGH: 0.05 mg/kg *5 d	Nitrogen retention was 2.4-fold greater in combination	NA	Combination of GH and IGF-I was substantially more anabolicthan either GH or IGF-I alone.
**Lagger**[69]	24 healthy male	Randomised and paired three groupsIGF-1-insulin	**High doses** (30 μg/kg/h IGF-I or 0.23 nmol/kg/h insulin);**low doses** (5 μg/kg/h IGF-I or 0.04 nmol/kg/h insulin)	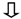 Ra 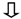 Rd------------Any change compared to control	NA	High doses of IGF-1 compared to insulin decreased more proteolysis. Unexpected inhibition of protein synthesis
**Rutter**[70]	Boys with DMD21 control17 IGF1	Randomised, rhIGF-1 vs. placebo.6 month	GC-treated + IGF-1 (n = 17) vs. controls (GC-therapy only n = 21)	NA	Did not observe a change in functional motor outcomes	Boys with DMD. 6 months of rhIGF-1 therapy did not change motor function but did improve linear growth.
**Nishan Guha**[71]	Fifty-six recreational athletes	Randomised, double-blind, placebo-controlled	Low dose rhIGF-I/rhIGFBP-3(30 mg/d), or high dose rhIGF-I/rhIGFBP-3 (60 mg/d) for 28 d	NA	Significant increase inmaximal oxygen consumption	rhIGF-I/rhIGFBP-3 administration for 28 days improved aerobic performance in recreationalathletes, with no effects on body composition.

DMD = Duchenne muscular dystrophy EA: essential amino acid IBW: ideal body weight. Ins: insulin, Leu: leucine, Ox: oxidation, BCAAs: branched-chain amino acids, Phe: phenylalanine, wk week. NA: not available. (*) during. Units: d. day, kg: kilogram, h: hours, IU: international units, min: minutes, mg: milligram, mU: primary units, μg: microgram nmol: nanomole pmol: picomole, Rd: (Phe/Leu disappearance rate or nonoxidative AA disposal) represents protein synthesis. Ra: (rate of appearance of Phe/Leu) represents proteolysis.

## Data Availability

Not applicable.

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
