# Peer review of "GH/IGF-1 Abnormalities and Muscle Impairment: From Basic Research to Clinical Practice"

_ijms, 2021, doi:10.3390/ijms22010415_

Round 1
Reviewer 1 Report
This article was dedicated to review the musculoskeletal symptoms and the impact of GH and IGF-1 in patients suffering from the acromegaly. The idea is good but the fulfillment is definitely not the best.
Major remarks:
- The structure of this review is more similar to the fact-gathering collection from the various articles than to a real review, since the concrete and clear summarizing conclusions are missing.
- For example, the final conclusions after all 15 pages of review says that: “…our knowledge about the underlying mechanisms and evolution after the different treatments is very limited”, “The degree of contribution of GH or IGF-1 to the myopathy in acromegaly remains to be further investigated”, “we need more studies, ….tools, …. collaborative, multicentre studies….”, “…more studies are needed…”. Not clear what “basic y clinical” (444 line) means as well as “muscle should be seen as double organ” (452 line). If the authors cannot do any conclusions from 127 cited sources, what is a meaning to write a review?
- The title says “…clinical features and underlying mechanisms”, howler, none of the mechanisms was highlighted. The title also suggests that review should be about the patients’ data, but there are animals and animal tissues-derived cells as well.
- The review, probably, would be more understandable if authors would concentrate only on the findings in human samples and would do a proper analysis of findings with as much as possible concrete conclusions from these findings, and would skip a basic research part (cell lines and transgenic mice), which also has many inaccuracies: C2C12 is a cell line, not a primary cell culture. Description of cell effects in 6.1. part very often is missing the source of the cells and more detail explanation.
- The review has very many inaccuracies and mistakes, the English language should be strongly checked and corrected.
It is impossible to mention all inaccuracies of English language, just some of them:
- The meaning of paragraph titles is difficult to follow, i.e. it is difficult to understand what authors will be talking about in the paragraphs: “Muscle in GH /IGF1 deficiency”, “GH/IGF-1 axis…”, “The skeletal muscle a central structural and metabolic organ”, “Muscle in Acromegaly” and many other.
- The 8th part “Muscle in Acromegaly” contains mostly human data and one mentioning of zebrafish. This part could be shortened to only findings in human.
- Tables 1 and 2 are total mess and chaos. The intervals are minimal between columns and impossible to follow information. The units and dimensions should be written with the intervals “0,014 μgr/kg/mi”, not like “0,014μgr/kg/mi”, which actually should be “……/min”. Arrows are everywhere and impossible to follow their real meaning. Units are also strange, such as: μgr, hs, mcg, wk and many other. Min-1 and other should be also written in superscript way. There is a lot what to correct in these tables.
- Legends of the Tables 1 and 2 have many not explained abbreviations. In addition, the abbreviations in whole review should be carefully checked, many of them are missing an extension.
- Many expressions are not clear at all, such as: “ …muscle is structural organ” (line 118); “when food scares” (165 line); “anerobic” (lien 133), “There are some differences among the various transgenic lines species (ovine, bovine or human)” (311 line), “…about the effects on muscle of GH/IGF-1 system” (341 line); “stablished” (396 line); “muscle should be seen as double organ” (452 line) and many other.
Author Response
List of responses to the reviewers’ comments:
Thank you very much for the review process and for giving us the opportunity to revise our manuscript. All the reviewers’ comments have been answered (red font) and the paper has been modified accordingly. We look forward to hearing from you in due time regarding our submission and to respond to any further questions and comments you may have.
Reviewer #1:
Comments and Suggestions for Authors
This article was dedicated to review the musculoskeletal symptoms and the impact of GH and IGF-1 in patients suffering from the acromegaly. The idea is good but the fulfillment is definitely not the best.
Major remarks:
- The structure of this review is more similar to the fact-gathering collection from the various articles than to a real review, since the concrete and clear summarizing conclusions are missing.
- For example, the final conclusions after all 15 pages of review says that: “…our knowledge about the underlying mechanisms and evolution after the different treatments is very limited”, “The degree of contribution of GH or IGF-1 to the myopathy in acromegaly remains to be further investigated”, “we need more studies, ….tools, …. collaborative, multicentre studies….”, “…more studies are needed…”. Not clear what “basic y clinical” (444 line) means as well as “muscle should be seen as double organ” (452 line). If the authors cannot do any conclusions from 127 cited sources, what is a meaning to write a review?
Answer: Many thanks for the revision process and the criticism of our paper. The referee is right in indicating that the concluding remarks should be improved. In this regard, the paragraphs have been structured and all this section has been improved in terms of clarity. However, we feel that it is also important to highlight the scientific gaps which, unfortunately, are significant in this research area.
- The title says “…clinical features and underlying mechanisms”, howler, none of the mechanisms was highlighted. The title also suggests that review should be about the patients’ data, but there are animals and animal tissues-derived cells as well.
Answer: The title has been changed according to the suggestions of the reviewer.
- The review, probably, would be more understandable if authors would concentrate only on the findings in human samples and would do a proper analysis of findings with as much as possible concrete conclusions from these findings, and would skip a basic research part (cell lines and transgenic mice), which also has many inaccuracies: C2C12 is a cell line, not a primary cell culture. Description of cell effects in 6.1. part very often is missing the source of the cells and more detail explanation.
Answer: Following the reviewer’s recommendation, we have emphasized the human findings (section 6.3 has been significantly shortened). However, we feel that some examples of basic research are helpful for the better understanding of the scientific gaps. The inaccuracies have been corrected, and more detailed explanations about the source of the cells have been added as required.
- The review has very many inaccuracies and mistakes, the English language should be strongly checked and corrected.
It is impossible to mention all inaccuracies of English language, just some of them:
Answer: The English has been corrected throughout the manuscript.
- The meaning of paragraph titles is difficult to follow, i.e. it is difficult to understand what authors will be talking about in the paragraphs: “Muscle in GH /IGF1 deficiency”, “GH/IGF-1 axis…”, “The skeletal muscle a central structural and metabolic organ”, “Muscle in Acromegaly” and many other.
Answer: Following the reviewer’s recommendation, the subtitles have been replaced by more informative ones.
7. The 8th part “Muscle in Acromegaly” contains mostly human data and one mentioning of zebrafish. This part could be shortened to only findings in human.
Answer: As recommended, the findings on zebrafish have been eliminated from the revised manuscript.
8. Tables 1 and 2 are total mess and chaos. The intervals are minimal between columns and impossible to follow information. The units and dimensions should be written with the intervals “0,014 μgr/kg/mi”, not like “0,014μgr/kg/mi”, which actually should be “……/min”. Arrows are everywhere and impossible to follow their real meaning. Units are also strange, such as: μgr, hs, mcg, wk and many other. Min-1 and other should be also written in superscript way. There is a lot what to correct in these tables.
Answer: Thank you for this important comment. It seems that there have been editing problems and we apologize for this issue. In the revised manuscript we have re-edited both tables and the meaning of the arrows and abbreviations has been added in the footnotes. We have also addressed the errors in units.
9. Legends of the Tables 1 and 2 have many not explained abbreviations. In addition, the abbreviations in whole review should be carefully checked, many of them are missing an extension.
Answer: We have carefully checked all manuscript and tables to spell out all the abbreviations.
10. Many expressions are not clear at all, such as: “ …muscle is structural organ” (line 118); “when food scares” (165 line); “anerobic” (lien 133), “There are some differences among the various transgenic lines species (ovine, bovine or human)” (311 line), “…about the effects on muscle of GH/IGF-1 system” (341 line); “stablished” (396 line); “muscle should be seen as double organ” (452 line) and many other.
Answer: The text has been English edited, and all spelling and grammatical errors pointed out and corrected (red font) in the revised manuscript
Reviewer 2 Report
The current manuscript arouses interest for readers and provides an important clue to understand the impact of GH/IGF-1 on skeletal muscle impairment in patients with skeletal muscle myopathy, including those with acromegaly. This manuscript has been well written and is easily readable to understand. However, there are many, many, many typographic and grammatical errors throughout this manuscript. Therefore, the authors must ask a native English speaker familiar with medicine to edit their paper, including figures, and also should proofread their manuscript more carefully by themselves.
Author Response
Thank you very much for the review process and for giving us the opportunity to revise our manuscript. All the reviewers’ comments have been answered (red font) and the paper has been modified accordingly. We look forward to hearing from you in due time regarding our submission and to respond to any further questions and comments you may have.
Reviewer #2
The current manuscript arouses interest for readers and provides an important clue to understand the impact of GH/IGF-1 on skeletal muscle impairment in patients with skeletal muscle myopathy, including those with acromegaly. This manuscript has been well written and is easily readable to understand. However, there are many, many, many typographic and grammatical errors throughout this manuscript. Therefore, the authors must ask a native English speaker familiar with medicine to edit their paper, including figures, and also should proofread their manuscript more carefully by themselves.
Answer: Many thanks for the revision process and your kind comments on our paper. The text has been English edited, and all spelling and grammatical errors corrected.
Round 2
Reviewer 1 Report
The review has been considerably improved.
Few last minor corrections and suggestions:
- It is better to use “µg” instead “µgr” since it is more accepted way in science.
- 314 line. Since authors are talking about one C2C12 myoblast line, the plural term should be changed.
- 339 line. Probably authors are talking about “transgenic cell lines” of various species, otherwise “transgenic human” so far cannot be...
- 426 line. Section 8 – better to write “…..in human Acromegaly” (if authors are talking only about patients).
Author Response
List of responses to the reviewer comments: (It is highlighted in yellow in the text)
1.It is better to use “µg” instead “µgr” since it is more accepted way in science.
Answer: We agree with this comment and, consequently, we have proceed as recommended.
2. 314 line. Since authors are talking about one C2C12 myoblast line, the plural term should be changed.
Answer: This has been corrected as suggested. Thank you!
3. 339 line. Probably authors are talking about “transgenic cell lines” of various species, otherwise “transgenic human” so far cannot be...
Answer: The referee is right and the sentence have modified as follows: “Various species of GH transgenes in mice (e.g. ovine GH, bovine GH, human GH) ”
4. 426 line. Section 8 – better to write “…..in human Acromegaly” (if authors are talking only about patients).
Answer: The title has been changed as suggested.